# Temporal Label-Refinement for Weakly-Supervised Audio-Visual Event Localization

## Abstract

Audio-Visual Event Localization (AVEL) is the task of temporally localizing and classifying *audio-visual events*, i.e., events simultaneously visible and audible in a video. In this paper, we solve AVEL in a weakly-supervised setting, where only video-level event labels (their presence/absence, but not their locations in time) are available as supervision for training. Our idea is to use a base model to estimate pseudo labels on the training data at a finer temporal resolution than at the video level ("label-refinement") and then re-train the model with these new labels. In label-refinement, we estimate the subset of labels for each *slice* of frames in a training video by (i) replacing the frames outside the slice with those from a second video having no overlap in video-level labels, and (ii) feeding this synthetic video into the base model to extract labels for just the slice in question. To handle the out-of-distribution nature of our synthetic videos, we propose an auxiliary objective to train the base model that induces more reliable predictions of the localized event labels as desired. Our three-stage pipeline outperforms several existing AVEL methods with no architectural changes and improves performance on a related weakly-supervised task as well. We also find that the evaluation of existing AVEL methods has been seriously misleading and therefore propose new metrics for a better sense of performance.

## 1 Introduction

A crucial milestone in bridging the gap between human and machine intelligence is to have machines jointly reason about the multiple modalities of information (e.g., visual, audio, and text) in the world. To this end, researchers have introduced various subproblems (Tian et al., 2018; 2020; Arandjelovic & Zisserman, 2017) in multimodal learning to drive innovation in the field. An important *joint* reasoning problem is the task of Audio-Visual Event Localization (AVEL) (Tian et al., 2018), illustrated in Fig. 1. Given a video, the objective is to temporally localize events that are both audible and visible at the same instant, i.e., *audio-visual events*, and classify them into a set of known event categories. Events/actions that are either audible or visible but *not both* (e.g., commentary during a televised football game) are not classified as audio-visual events. For a network to perform well at such a task, it needs to implicitly learn to combine information from the two modalities at each instant and determine whether they correspond or not.

Some of the most notable advances in deep learning (He et al., 2016; Brown et al., 2020) have stemmed from access to large-scale datasets. Large-scale, fully annotated datasets for videos would require watching and listening to hundreds of thousands of videos and manually labeling each frame in each video. *Weakly-supervised learning* (learning from underspecified labels) aims to alleviate this cost. In our context, weak supervision is the scenario where only the *set* of audio-visual events occurring in a video is available for that video in the training data (we are still required to temporally localize events in the test phase).

In this paper, we present a novel method to solve AVEL in a weakly-supervised setting. While much progress (Zhou et al., 2021; Xuan et al., 2020; Lin et al., 2019; Ramaswamy & Das, 2020; Ramaswamy, 2020; Xu et al., 2020; Lin & Wang, 2020) has been made for weakly-supervised AVEL since the pioneering work of Tian et al. (2018), this has mainly taken the form of architectural and feature-aggregation modifications (see Sec. 2). Different from these approaches, however, we fix the network architecture to that of the very

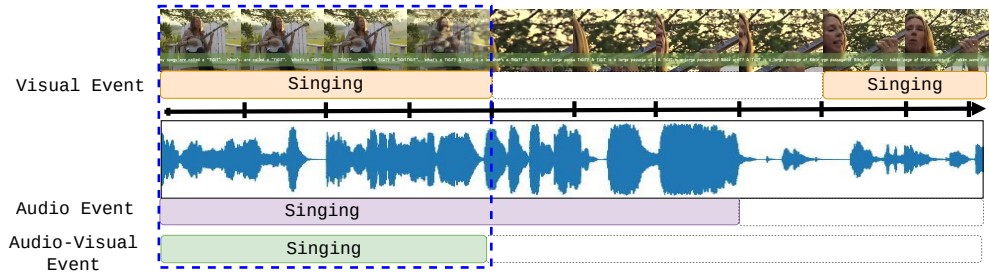

Figure 1: The AVEL task. The event "Singing" occurs during $[0, 4]$ seconds and $[8, 10]$ seconds in the visual modality. It also occurs during $[0, 7]$ seconds in the audio modality. However, only the segments where it occurs in *both* modalities are labeled audio-visual events (AVEs). In this case, the AVE "Singing" is said to occur during $[0, 4]$ seconds (see blue dashed lines).

first baseline for AVEL (Tian et al., 2018) and show how to exploit its existing predictive power with a carefully designed training strategy that yields significant performance gains. Moreover, while architectural changes may constrain a method to the task at hand, better training strategies could potentially generalize to related tasks. E.g., our method is easily extended to enhance performance on the more challenging weakly-supervised Audio-Visual Video Parsing (AVVP) (Tian et al., 2020) task. Our key idea is to create a middle-ground between the fully- and weakly-supervised settings by employing a base model to estimate pseudo-labels on the training data that are more localized in time than at just the video level. We achieve this by feeding special synthetic videos to the trained base model and extracting its video-level predictions. Since the out-of-distribution (OOD) nature of our synthetic videos w.r.t. the base model could lead to unreliable estimates, we design an auxiliary training objective for training base model that *prepares* it to handle such OOD inputs. Finally, we re-train the base model with the refined labels.

## 2   Related Work

**Weakly-Supervised Event Localization in Videos.** Several methods (Islam et al., 2021; Luo et al., 2020; Nguyen et al., 2018; Shi et al., 2020; Wang et al., 2017) have been proposed for weakly-supervised Temporal Action Localization (TAL), which aims to classify and localize *visual* events in videos. For the more challenging AVEL task, existing methods have mainly focused on better audio-visual feature aggregation. Tian et al. (2018) proposed Audio-Guided Visual Attention (AGVA) to select visual features that correspond most to the audio. Lin et al. (2019) processed local and global audio-visual features with an LSTM-based network. Xuan et al. (2020) proposed spatial and temporal attention mechanisms to select the most discriminative event-related information. Similarly, Lin & Wang (2020) proposed an audio-visual transformer module that aggregates relevant intra- and inter-frame visual information. Ramaswamy (2020) explored audio-visual feature fusion methods to capture intra- and cross-modal relations. Zhou et al. (2021) constructed an all-pair audio-visual similarity matrix to inform feature aggregation across video frames.

**Audio-Visual Video Parsing (AVVP)** (Tian et al., 2020) aims at labeling events in a video as audible/visible/both, as well as temporally localizing and classifying them under weak supervision. Tian et al. (2020) formulated AVVP as a Multimodal Multiple-Instance Learning (MMIL) problem and proposed a hybrid attention network to capture unimodal and cross-modal contexts. Our train-infer-retrain pipeline was inspired by Wu & Yang (2021), who inferred modality-aware labels (`MA`) for AVVP by exchanging the audio/visual streams between pairs of videos. However, we note crucial differences: (i) We refine labels along the temporal axis with a sliding window instead of estimating them for an entire modality. Re-training with such labels does not follow from `MA` since their labels are not localized in time. (ii) `MA` could not effectively localize events in time without a separate contrastive loss, meaning temporal refinement is not a trivial extension. (iii) Our synthetic videos are discontinuous in time while theirs remain coherent, and our auxiliary objective helps the base model maintain reliable predictions for such videos. (iv) Unlike `MA`, which specifically solves AVVP, our method applies to AVEL and AVVP and might inspire weakly-supervised methods more generally.

**Pseudo-Labeling** refers to estimating labels for unlabeled data using the predictions of a trained model. Pseudo-labeling has been used to improve performance on several weakly-supervised tasks including object detection (Tang et al., 2017; Zhou et al., 2016; Bilen & Vedaldi, 2016) and image classification (Ge et al., 2019; Cabral et al., 2014; Hu et al., 2019). A few works (Zhai et al., 2020; Luo et al., 2020; Pardo et al., 2021) have employed pseudo-labeling to improve performance on Temporal Action Localization (TAL), generating labels from model outputs or attention weights.

## 3 Problem Definition

**Preliminaries.** In the AVEL problem, an input video $V$ is partitioned into a set of $T$ non-overlapping (but contiguous) temporal segments $\{(S_t^v, S_t^a)\}_{t=1}^T$, where $S^v$ and $S^a$ are the visual and audio streams, respectively. Each segment is 1s long, and the number of segments $T$ is the same across videos. Given a video, the objective is to classify each segment $(S_t^v, S_t^a)$ into one of $C+1$ classes, where the first $C$ represent audio-visual events (e.g., "man speaking", "violin", etc., that are simultaneously visible and audible in the segment). The last class is *background*, which applies when the event occurring in the segment is either visible or audible but not both (or when it does not belong to any of the first $C$). We denote each segment-level label by a one-hot vector $\mathbf{y_t} \in \{0,1\}^{C+1}$, where $\sum_{c=1}^{C+1} y_t(c) = 1$.

**Weak-Supervision.** In the weakly-supervised setting, we do not have access to the segment-level labels $\{\mathbf{y_t}\}_{t=1}^T$ for training. For each training video, we are instead provided with a video-level label $\mathbf{Y} \in \{0,1\}^{C+1}$ that indicates only the presence/absence of audio-visual events in the video, but not their locations in time. Note that $Y(C+1) = 1$ if no segment in the video contains an audio-visual event. For $c \in [1, C]$, $Y(c) = 1$ if *some* segment contains that audio-visual event.

Some prior work (Lin et al., 2019; Xuan et al., 2020; Zhou et al., 2021) has adopted an alternative definition of the weakly-supervised setting, where the weak labels for training are taken as $\mathbf{Y} = \frac{1}{T}\sum_{t=1}^T \mathbf{y_t} \in [0,1]^{C+1}$. I.e., they assume access to not just the set of audio-visual events in a video but also the durations (not locations) for which they occur. E.g., if $Y(c) = 0.9$ for some $c \in [1, C]$, then that event must have occurred in almost all (90% of) segments in the video. Weak labels of this form encode more information than in the original formulation. However, it is no easier to collect such labeled data than it is to collect a fully-annotated dataset. We, therefore, adhere to the original weakly-supervised formulation in our experiments and comparisons with prior work.

## 4 Method

### 4.1 Base Model Architecture

Since our objective is to improve weakly-supervised performance without relying on architectural changes, we follow the baseline architecture from Tian et al. (2018), outlined below.

**Feature Extraction.** For each video segment, pre-trained CNNs $\mathbf{\Phi^v}$ and $\mathbf{\Phi^a}$ extract visual and audio representations, $\mathbf{f_t^v} = \mathbf{\Phi^v}(S_t^v) \in \mathbb{R}^{w^2 \times n_c}$ and $\mathbf{f_t^a} = \mathbf{\Phi^a}(S_t^a) \in \mathbb{R}^{n_a}$, respectively. Here, $w$ is the spatial dimension of the output of the CNN layer, $n_c$ is the number of channels, and $n_a$ is the dimension of the audio feature.

**Audio-Guided Visual Attention.** This aims to exploit the natural correspondence between audio and video signals to allow the former to inform the network about the most relevant image regions that correspond to it. The visual features representing these regions are then weighted favorably in feature aggregation. I.e., each visual segment is represented with the spatial-aggregate $\mathbf{f_t^{v,att}} = \sum_{k=1}^{w^2} \alpha_t(k)\mathbf{f_t^v(k)} \in \mathbb{R}^{n_c}$, where the attention weights $\boldsymbol{\alpha_t}$ are inferred for the segment as:

$$\mathbf{z_t} = U^v(\mathbf{f_t^v})\mathbf{W^v} + \mathbf{W^a}U^a(\mathbf{f_t^a})\mathbf{1}^T \in \mathbb{R}^{w^2 \times d}$$

$$\boldsymbol{\alpha_t} = \text{SoftMax}(\tanh(\mathbf{z_t})\mathbf{W^f}) \in [0,1]^{w^2}, \tag{1}$$

where $U^v : \mathbb{R}^{w^2 \times n_c} \mapsto \mathbb{R}^{w^2 \times h}$ and $U^a : \mathbb{R}^{n_a} \mapsto \mathbb{R}^h$ are fully-connected (ReLU) layers, $\mathbf{W^v} \in \mathbb{R}^{h \times d}$, $\mathbf{W^a} \in \mathbb{R}^{w^2 \times h}$, and $\mathbf{W^f} \in \mathbb{R}^d$ are learnable projection matrices, and $\mathbf{1} \in \{1\}^d$.

**Temporal Modeling.** Temporal context from neighboring segments is incorporated into the visual and audio features $\mathbf{f_t^{v,att}}$ and $\mathbf{f_t^a}$, respectively, using separate bi-directional LSTMs (Hochreiter & Schmidhuber, 1997):

$$\{\mathbf{h_t^v}\}_{t=1}^T = \text{Bi-LSTM}^v(\{\mathbf{f_t^{v,att}}\}_{t=1}^T) \tag{2}$$

$$\{\mathbf{h_t^a}\}_{t=1}^T = \text{Bi-LSTM}^a(\{\mathbf{f_t^a}\}_{t=1}^T), \tag{3}$$

where $\mathbf{h_t^v} \in \mathbb{R}^{2h}$ and $\mathbf{h_t^a} \in \mathbb{R}^{2h}$ are the hidden states of the two LSTMs.

**Multimodal Fusion.** The resulting segment-level visual and audio representations are concatenated along the feature dimension to obtain:

$$\mathbf{h_t^*} = \text{Concat}[\mathbf{h_t^v}; \mathbf{h_t^a}] \in \mathbb{R}^{4h}. \tag{4}$$

**MIL and Classification.** The fused features are first transformed into raw segment-level class scores

$$\mathbf{x_t} = \mathbf{W^o}U^o(\mathbf{h_t^*}) \in \mathbb{R}^{C+1}, \tag{5}$$

where $U^o : \mathbb{R}^{4h} \mapsto \mathbb{R}^{h'}$ is a fully-connected (ReLU) layer, and $\mathbf{W^o} \in \mathbb{R}^{(C+1) \times h'}$ is a learnable projection matrix. Finally, Multiple-Instance Learning (Dietterich et al., 1997) (MIL) is used to train the base model with the weak labels provided. The video-level prediction $\hat{\mathbf{Y}}$ is computed as:

$$\hat{\mathbf{Y}} = \text{SoftMax}(\text{MaxPool}(\{\mathbf{x_t}\}_{t=1}^T)) \in [0, 1]^{C+1}. \tag{6}$$

$\hat{\mathbf{Y}}$ is optimized to match $\mathbf{Y}$ with the multi-class soft margin loss function. During inference, segment-level predictions are obtained by finding the largest entry in $\mathbf{x_t}$.

## 4.2 Temporal Label-Refinement

Our goal here is to estimate event labels for *slices* (windows) of segments in training videos and then re-train the base model using the derived labels. "Slice" here means $N$ consecutive segments in a video ($N < T$).

**Notation.** Consider the $i$-th training video $V^{(i)}$. Let $L^{(i)}$ be the set of audio-visual events (if any) occurring in the video. Let $L^{(i)}[t_1, t_2]$ be the set of audio-visual events occurring within segments $[t_1, t_2]$ (both segments included), where $1 \leq t_1 \leq t_2 \leq T$, and $t_1, t_2 \in \mathbb{N}$. Finally, let $[t_1, t_2]^c$ be the complementary duration $[1, t_1 - 1] \cup [t_2 + 1, T]$, and $L^{(i)}[t_1, t_2]^c$ be the set of audio-visual events occurring in this duration. Clearly, $L^{(i)}[t_1, t_2] \subseteq L^{(i)}$ and $L^{(i)}[t_1, t_2]^c \subseteq L^{(i)}$.

**Motivation.** In terms of this notation, the weakly-supervised setting can be described as having access to $L^{(i)} = L^{(i)}[1, T]$ during training. The fully-supervised setting, on the other hand, allows access to $L^{(i)}[t, t]$ for each $t$ in $V^{(i)}$, and is the ideal training scenario for model performance. We seek to create a middle-ground setting where we have access to $L^{(i)}[t, t + N - 1]$ for training, with $1 < N \ll T$ and $N \in \mathbb{N}$. We will achieve this by merely exploiting the ability of our base model (Sec. 4.1) in making *video-level* predictions, a task it was directly trained for.

Note that such labels are more informative than labels at the video level because they are localized over a shorter duration ($N$ as opposed to $T$s) in the video. E.g., the audio-visual event of a church bell ringing may only last for the first 3 segments in a video (out of, say, $T = 10$), after which it stops ringing and is no longer audible. With localized labels (say $N = 5$), this event would be included in $L^{(i)}[1, 5]$ but *not* in $L^{(i)}[4, 8]$. In the absence of such labels, the event would be included in $L^{(i)}[1, 10]$, with no extra information about its extent in time. Thus, localized labels, once obtained, would provide stronger supervision in the training phase.

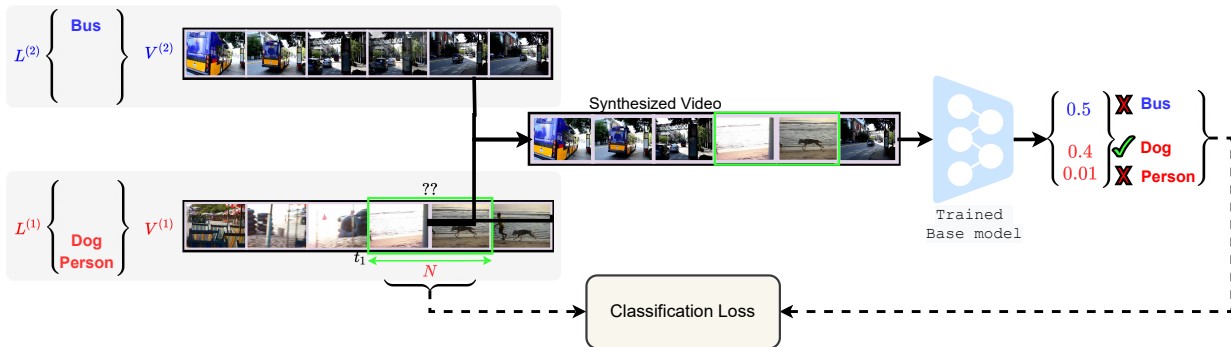

Figure 2: The "label-refinement" (slice-level pseudo-label extraction) method. We start with two training videos $V^{(1)}$ and $V^{(2)}$ with no common events and wish to determine *localized* labels for the $N$-segment window in $V^{(1)}$. We feed the synthetic video into a trained base model and extract the *video-level* predictions on the right. The event "Bus" receives a probability of 0.5 in the synthetic video but could not have occurred anywhere in $V^{(1)}$, which we know from access to the video-level labels $L^{(1)} = \{\text{Dog}, \text{Person}\}$. "Person" does occur in $V^{(1)}$ but receives a very low probability (0.01) in the synthetic video. Therefore, the estimated answer to '??' is just the set {"Dog"}. The dashed lines mean the model is re-trained to predict {"Dog"} for this window in the next training stage. We repeat this for all slices in $V^{(1)}$, moving at a stride $s \geq 1$.

**Method.** Consider two training videos $V^{(i)}$ and $V^{(j)}$, and their corresponding label sets $L^{(i)}$ and $L^{(j)}$. We have:

$$L^{(i)} \cap \left( L^{(i)}[t_1, t_2] \cup L^{(j)}[t_1, t_2]^c \right) = \left( L^{(i)} \cap L^{(i)}[t_1, t_2] \right) \cup \left( L^{(i)} \cap L^{(j)}[t_1, t_2]^c \right)$$
$$= L^{(i)}[t_1, t_2] \cup \left( L^{(i)} \cap L^{(j)}[t_1, t_2]^c \right). \tag{7}$$

Now, assume that $V^{(j)}$ is chosen such that it has no overlap in video-level labels with $V^{(i)}$, i.e., $L^{(i)} \cap L^{(j)} = \emptyset$. Since $L^{(j)}[t_1, t_2]^c \subseteq L^{(j)}$, Eq. (7) reduces to:

$$L^{(i)}[t_1, t_2] = L^{(i)} \cap \underbrace{\left( L^{(i)}[t_1, t_2] \cup L^{(j)}[t_1, t_2]^c \right)}_{\text{Term (*)}}. \tag{8}$$

This suggests a way to obtain the localized labels $L^{(i)}[t_1, t_2]$ we seek. $L^{(i)}$ is available in the training data. Term (*) represents the union of audio-visual events occurring in $[t_1, t_2]$ from $V^{(i)}$ and in $[t_1, t_2]^c$ from $V^{(j)}$. In other words, if we synthesize a video $\tilde{V}^{(i)}$ by retaining the segments in $[t_1, t_2]$ from $V^{(i)}$ and replacing the rest with those taken from $V^{(j)}$, Term (*) would represent the set of video-level labels for $\tilde{V}^{(i)}$. Since we already have a base model trained under weak supervision to make video-level predictions, it could be expected to act as an accurate *video-level* predictor (but not as a *segment-level* predictor). So, we feed our synthetic video $\tilde{V}^{(i)}$ to the base model to obtain an estimate of Term (*), and then use Eq. (8) to estimate $L^{(i)}[t_1, t_2]$ as desired.

**Implementation Steps:**

**1. Training a Base Model.** We first train the model ($\mathbf{\Phi_0}$) described in Sec. 4.1 in the weakly-supervised setting, i.e., $\hat{\mathbf{Y}}^{(i)} = \mathbf{\Phi_0}(V^{(i)}) \in [0, 1]^{C+1}$.

**2. Label-Refinement.** For each training video $V^{(i)}$, we randomly sample a second video $V^{(j)}$ having no common labels with $V^{(i)}$. We fix a window size of $N$ segments and proceed in a *sliding-window* fashion to estimate $L^{(i)}[t_1, t_1 + N - 1]$ for each permissible starting segment $t_1$ as described below.

First, we create the synthetic video by taking $V^{(i)}$ and replacing its segments outside $[t_1, t_1 + N - 1]$ with the corresponding segments from $V^{(j)}$. We call the resulting video $\tilde{V}^{(i;t_1)}$, computed as follows:

$$\tilde{V}^{(i;t_1)}[1, T] := V^{(i)}[1, T]$$
$$\tilde{V}^{(i;t_1)}[t_1, t_1 + N - 1]^c = V^{(j)}[t_1, t_1 + N - 1]^c. \tag{9}$$

Next, we use $\boldsymbol{\Phi_0}$ to make *video-level* predictions on $\tilde{V}^{(i;t_1)}$ and filter them with the video-level label vector $\mathbf{Y^{(i)}}$ ($\odot$ is the element-wise product), following Eq. (8):

$$\hat{\mathbf{Z}}^{(i;t_1)} = \mathbf{Y^{(i)}} \odot \boldsymbol{\Phi_0}(\tilde{V}^{(i;t_1)}) \in [0, 1]^{C+1}. \tag{10}$$

Finally, we estimate $L^{(i)}[t_1, t_1 + N - 1]$ using an event-detection threshold $\tau \in (0, 1)$ as follows:

$$L^{(i)}[t_1, t_1 + N - 1] = \{c \in [1, C] \mid \hat{Z}^{(i;t_1)}(c) \geq \tau\}. \tag{11}$$

We repeat this for all slices by moving the starting location, $t_1$, forward at a fixed stride $s \geq 1$ such that $s | (T - N)$. We store all the slice-level pseudo-labels for each video in the training data for re-training.

**3. Re-training with Refined Labels.** Once we have obtained localized labels for all training videos, we re-train the base architecture under this more strongly-supervised setting. This is straightforward because the MIL pooling operation (Sec. 4.1) is indifferent to the number of instances taken in a bag. Specifically, we first feed $V^{(i)}$ into the base architecture and extract the raw segment-level class scores $\{\mathbf{x_t^{(i)}}\}_{t=1}^{T}$. Representing each estimated $L^{(i)}[t_1, t_1 + N - 1]$ as a vector $\mathbf{Y^{(i;t_1)}} \in \{0, 1\}^{C+1}$, we calculate the label-refinement loss $\mathcal{L}_{\mathrm{LR}}$ for $V^{(i)}$ as:

$$\hat{\mathbf{Y}}^{(i;t_1)} = \mathrm{SoftMax}(\mathrm{Pool}(\{\mathbf{x_t^{(i)}}\}_{t=t_1}^{t_1+N-1})) \tag{12}$$
$$\mathcal{L}_{\mathrm{LR}} = \frac{1}{T_1} \sum_{t_1} g\left(\hat{\mathbf{Y}}^{(i;t_1)}, \mathbf{Y^{(i;t_1)}}\right), \tag{13}$$

where $g$ is the classifier loss function applied in the weakly-supervised setting, i.e., $\mathcal{L}_{\mathrm{MIL}} = g(\hat{\mathbf{Y}}^{(i)}, \mathbf{Y^{(i)}})$, $t_1 \in \{1, 1 + s, 1 + 2s, ..., T - N + 1\}$ is the starting location of the window, and $T_1$ is the number of window locations permissible. We re-train from scratch imposing this additional refinement loss (averaged over training examples). Note that MIL pooling is now performed over $N$ instances as opposed to all $T$ instances originally. At test time, we use segment-level predictions as usual. Thus, our method lets us exploit the base model's video-level predictive ability to estimate labels for training videos that are more localized in time. Fig. 2 illustrates the label-refinement idea.

## 4.3 Auxiliary Training Objective

**Motivation.** One caveat with the label-refinement approach is that the synthetic videos $\tilde{V}^{(i)}$ do not belong to the distribution of examples $V^{(i)}$ used to train the base model in Step1. Replacing segments introduces temporal discontinuities in the input that did not exist in the original training data. Moreover, by replacing most segments in $V^{(1)}$ with segments from $V^{(2)}$ (see Fig. 2), the synthetic video is dominated by the events in $V^{(2)}$. When such videos are passed into the base model to obtain video-level predictions in Step2, it may lose confidence in the events occurring in the few retained segments from $V^{(1)}$, leading to false negatives in the refined labels for $V^{(1)}$.

**Method.** We propose to mitigate this by encouraging the base model (during Step1) to maintain the audio-visual information from the retained segments when faced with the new information from the second video. Recall from Sec. 4.2 that the video-level labels for the synthetic video $\tilde{V}^{(i;t_1)}$ are given by $L^{(i)}[t_1, t_1 + N - 1] \cup L^{(j)}[t_1, t_1 + N - 1]^c$. Our main idea is that with appropriate choices for the window size $N$ and stride $s$, the following relation holds:

$$\bigcup_{t_1} L^{(i)}[t_1, t_1 + N - 1] \cup L^{(j)}[t_1, t_1 + N - 1]^c = L^{(i)} \cup L^{(j)}. \tag{14}$$

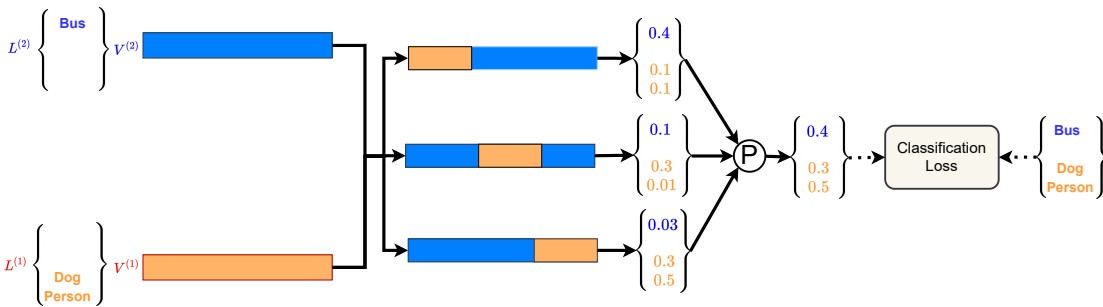

Figure 3: A schematic of the proposed auxiliary training objective. We start with two training videos with no common events. We synthesize three new videos as shown and feed each into the base architecture to extract raw score predictions. The scores are then pooled and optimized to predict the video-level labels $L^{(1)} \cup L^{(2)}$, encouraging the model not to ignore the events in $V^{(1)}$.

To see this, note that at the first location ($t_1 = 1$), the refinement window extends up to segment $N$ in $V^{(i)}$. At its final location, it extends back up to segment $T - N + 1$. Thus, $[t_1, t_1 + N - 1]^c$ will cover every segment of $V^{(j)}$ as long as $T - N + 1 > N$, or $N < \frac{T+1}{2}$.

Since $L^{(i)} \subseteq L^{(i)} \cup L^{(j)}$, we can encourage information retention for $V^{(i)}$ under segment replacement by aggregating the base model's outputs on the $T_1$ possible synthetic combinations $\{\tilde{V}^{(i;t_1)}\}_{t_1}$ for $V^{(i)}$, and optimizing the aggregate to predict *all* the labels in $L^{(i)} \cup L^{(j)}$, which we have access to in the training set. In other words, our objective should impose that when the video-level predictions for different synthetic videos are combined, *all* the events in $V^{(i)}$ (and $V^{(j)}$) can be recovered.

**Implementation.** The above idea is simple to incorporate into Step1, and *prepares* the base model for Step2. We randomly create synthetic videos as described and feed each video $\tilde{V}^{(i;t_1)}$ into the base architecture to extract the raw segment-level class scores $\{\tilde{\mathbf{x}}_{\mathbf{t}}^{(\mathbf{i};\mathbf{t_1})}\}_{t=1}^T$. We MIL-Pool these raw scores *within* and then *across* the $T_1$ synthetic videos as follows:

$$\tilde{\mathbf{x}}^{(\mathbf{i};\mathbf{t_1})} = \text{Pool}(\{\tilde{\mathbf{x}}_{\mathbf{t}}^{(\mathbf{i};\mathbf{t_1})}\}_{t=1}^T) \tag{15}$$

$$\tilde{\mathbf{x}}^{(\mathbf{i})} = \text{Pool}(\{\tilde{\mathbf{x}}^{(\mathbf{i};\mathbf{t_1})}\}_{t_1}) \in \mathbb{R}^{C+1}. \tag{16}$$

Finally, we generate an aggregate prediction $\tilde{\mathbf{Y}}^{(\mathbf{i})}$ and optimize it w.r.t. $\mathbf{Y}^{(\mathbf{ij})} \in \{0,1\}^{C+1}$, representing $L^{(i)} \cup L^{(j)}$. So, our auxiliary loss is computed as:

$$\tilde{\mathbf{Y}}^{(\mathbf{i})} = \text{SoftMax}(\tilde{\mathbf{x}}^{(\mathbf{i})}) \tag{17}$$

$$\mathcal{L}_{\text{A}} = g\left(\tilde{\mathbf{Y}}^{(\mathbf{i})}, \mathbf{Y}^{(\mathbf{ij})}\right), \tag{18}$$

and is added to $\mathcal{L}_{\text{MIL}}$ while training the base model in Step1. Fig. 3 illustrates the idea.

**Comparison with Mixup.** Our auxiliary objective has some similarity with Mixup regularization (Zhang et al., 2018), which implements the prior (for an image classifier) that a *convex sum* of inputs be predicted as a convex sum of labels. In contrast, we implement the prior that across all *convex concatenations* of inputs, *all* individual labels are recoverable by the base model (see Eq. (14)). Like Mixup, we perform data augmentation with synthetic examples when training the base model in Step1, preparing it for inference on out-of-distribution examples in Step2.

## 5 Experiments

### 5.1 Experimental Setup

**Dataset.** We use the publicly available AVE dataset collected by Tian et al. (2018). It contains 4143 10s-long videos ($T = 10$) with a train/val/test split of 3339/402/402 videos. Each video consists of a *single*

Table 1: Performance metrics for naive prediction strategies. '-' means the value is undefined. '*' indicates reproduced performance. All numbers are percentages.

| Method | Accuracy | Non-AVE F1 (R/P) |
|---|---|---|
| AVE (Tian et al., 2018)* | 67.1 | 2.1 (1.1/25.8) |
| AVE-repeat | 69.1 | - (0.0/-) |
| GT-repeat | 82.2 | - (0.0/-) |

audio-visual event belonging to one of $C = 28$ categories and each event is at least 2s long. There are an additional 178 videos that contain no audio-visual events.

**Evaluation Metrics.** So far, the only metric (Tian et al., 2018; Xuan et al., 2020; Lin et al., 2019; Ramaswamy & Das, 2020; Zhou et al., 2021; Ramaswamy, 2020; Xu et al., 2020; Lin & Wang, 2020) to evaluate AVEL performance has been segment-level classification accuracy. We argue that accuracy on its own is a seriously misleading performance metric for AVEL. To see this, we report the accuracies achieved by some naive prediction strategies in Tab. 1. We also report their F1 scores (along with recall/precision) in detecting *non*-audio-visual events (the background class). Here, AVE represents the base model trained in Step1. In AVE-repeat, we take the audio-visual class with the highest predicted *video-level* probability and repeat this prediction across all 10 segments. In GT-repeat, we take the ground truth *video-level* audio-visual event and repeat this prediction across all 10 segments. GT-repeat has an accuracy of 82.2% despite never having predicted a non-AVE correctly (0% non-AVE recall). This means only a minority (17.8%) of all segments in the AVE dataset do not contain audio-visual events. Consequently, AVE-repeat outperforms AVE in terms of accuracy but suffers in terms of non-AVE recall. Note that AVE itself achieves a relatively low non-AVE F1 score of 2.1%.

To summarize, a network could achieve high accuracy on the dataset by simply treating AVEL as a *video-level* classification problem as opposed to an event-localization problem. In Sec. 5.2, we show that previous methods indeed achieve high accuracies by predicting the same events everywhere. Therefore, we report all of the following segment-level metrics in our performance evaluations to get a better sense of model performance: (i) accuracy, (ii) overall (weighted) F1 score, (iii) F1 score in detecting non-AVE segments, and (iv) F1 score in classifying audio-visual events.

**Implementation Details.** We use VGG-19 (Simonyan & Zisserman, 2015) pre-trained on ImageNet as the visual feature extractor $\mathbf{\Phi^v}$. 16 video frames are sampled per second and their features are averaged to obtain a single representation per segment. We use a VGG-like (Hershey et al., 2017) network pre-trained on AudioSet (Gemmeke et al., 2017) as the audio feature extractor $\mathbf{\Phi^a}$. Each 1s audio is transformed into a log-Mel spectrogram before being fed into the network. We use the Adam optimizer (Kingma & Ba, 2015) with a batch size of 64 and a learning rate of 0.001 and train Step1 for 200 epochs and Step3 for 100 epochs to prevent overfitting. We take the detection threshold $\tau = 0.05$ in Eq. (11), based on performance on the validation set. For $N$ and $s$, we require $N < \frac{T+1}{2}$ (Sec. 4.3), want a roughly equal coverage of segments, and for faster training, only a few forward passes $T_1 = (T - N)/s + 1$ per video. Within these constraints, we find the best values, $N = 4$ and $s = 2$, using the validation set. More details are in Appendix A.2. Finally, because the AVE dataset has videos containing no audio-visual events, we conveniently sample the second video $V^{(j)}$ from these since $L^{(i)} \cap L^{(j)} = \emptyset$ holds trivially for any $V^{(i)}$ considered. Note that these videos still contain valid events in their audio and visual streams, but they do not co-occur.

## 5.2 Comparisons with Existing Methods

To make fair comparisons, we ensure that all methods considered (i) use the same pre-trained visual and audio feature extractors, (ii) are trained under the weakly-supervised setting opted for in Sec. 3, and (iii) are trained, validated, and tested on the train/val/test split provided in the AVE dataset.

**Quantitative.** We compare methods in Tab. 2 on all the segment-level performance metrics listed earlier. We report only the accuracy wherever the code bases are not publicly available. We can see that our method outperforms several existing methods on accuracy, overall F1 score, and non-AVE detection F1

Table 2: Performance comparison with previous methods under the weakly-supervised setting opted for in Sec. 3. '*' indicates reproduced performance. '-' means the code is not publicly available. Ours (XYZ) means we are using XYZ as the base model for our method.

| Method | Accuracy | Wt. F1 | Non-AVE F1 (R/P) | AVE F1 (R/P) |
|---|---|---|---|---|
| AVE* (Tian et al., 2018) | 67.1 | 61.5 | 2.1 (1.1/25.8) | 73.7 (81.3/67.4) |
| CMAN (Xuan et al., 2020) | 67.8 | - | - | - |
| AVSDN (Lin et al., 2019) | 68.4 | - | - | - |
| AVFB (Ramaswamy & Das, 2020) | 68.9 | - | - | - |
| AVIN (Ramaswamy, 2020) | 69.4 | - | - | - |
| CMRA* (Xu et al., 2020) | 69.6 | 63.5 | 0.5 (0.3/8.3) | 76.6 (84.6/70.0) |
| PSP* (Zhou et al., 2021) | 70.0 | 64.6 | 5.9 (3.3/29.3) | 77.2 (84.7/70.9) |
| AVT (Lin & Wang, 2020) | 70.2 | - | - | - |
| **Ours (AVE)** | **70.2** | **68.6** | **32.3** (25.5/43.9) | 76.4 (79.9/73.2) |

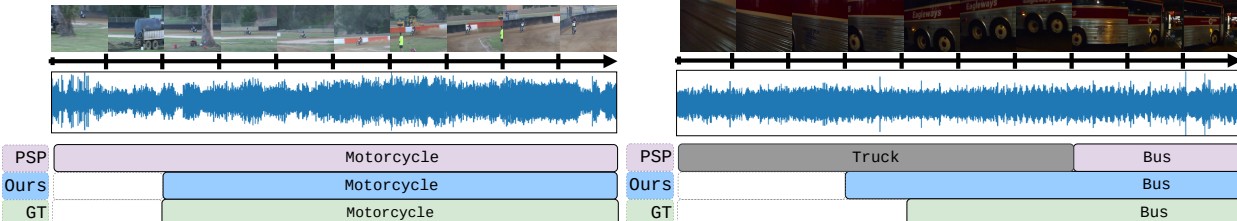

Figure 4: Qualitative comparison with PSP on two videos. "GT" is the ground truth.

score. In particular, we achieve an improvement of 26.4 points over PSP (Zhou et al., 2021), a state-of-the-art architecture, on non-AVE detection, with significantly higher recall and precision. Thus, our method can effectively discern the instances when the audio and visual signals are synchronized in a video leading to audio-visual events, and does not merely perform video classification, as discussed in Sec. 5.1. Moreover, since we use the same architecture as AVE and yet significantly outperform it, our results highlight the potential of training strategies to improve performance in the weakly-supervised setting.

**Qualitative.** We qualitatively compare the performance of our method (with an AVE base model) with PSP in Fig. 4. Previous methods often assign the predicted categories to *every* segment, despite achieving high accuracies on the dataset, as discussed in Sec. 5.1. On the other hand, our method classifies *and* localizes events more accurately. Appendix A.3 shows ten more examples.

### 5.3 Ablations

We report an ablation study in Tab. 3 to assess our proposed components. Here, AVE represents the base model trained in Step1. AVE+LR represents the model trained in Step3 with the refined labels obtained in Step2. To validate the effectiveness of label-refinement in determining the correct subset for each window, we train a model with the loss defined in Eq. (13), where we take $L^{(i)}[t_1, t_1 + N - 1] \equiv L^{(i)}$ everywhere. We call this AVE+LR$_{dummy}$. Finally, AVE+A+LR represents the three-step approach that includes the auxiliary objective in Step1 and re-training in Step3.

As expected, AVE+A+LR outperforms AVE+LR on all metrics. In particular, we get a near 5-point improvement in non-AVE precision, supporting our hypothesis in Sec. 4.3– there is now a decreased tendency of Step2 to lose confidence in legitimate audio-visual events (in the retained segments), reducing false-negative predictions on the training data. Similarly, AVE+LR outperforms AVE on all metrics, with a considerable improvement in non-AVE detection. It is interesting that while AVE+LR$_{dummy}$ outperforms AVE on accuracy, it is worse at determining whether a segment contains an audio-visual event or not. This is in line with our discussion in Sec. 5.1– encouraging each window to predict the global label $L^{(i)}$ is akin to solving *video classification.*

Table 3: Ablation study for our proposed Label-Refinement (LR) and Auxiliary Objective (A) ideas.

| Method | Accuracy | Wt. F1 | Non-AVE F1 (R/P) | AVE F1 (R/P) |
|---|---|---|---|---|
| AVE | 67.1 | 61.5 | 2.1 (1.1/25.8) | 73.7 (81.3/67.4) |
| AVE+PL | 67.5 | 62.3 | 3.2 (1.7/23.5) | 74.3 (81.7/68.1) |
| AVE+LR$_{dummy}$ | 67.8 | 61.9 | 0.5 (0.3/6.9) | 74.6 (82.4/68.2) |
| AVE+LR | 69.1 | 67.7 | 30.7 (25.3/39.1) | 75.7 (78.6/73.0) |
| **AVE+A+LR** | **70.2** | **68.6** | **32.3** (25.5/43.9) | **76.4** (79.9/73.2) |

Table 4: Impact of $\tau$ on AVE+A+LR accuracy.

| $\tau$ | 0.01 | 0.03 | 0.05 | 0.07 | 0.10 |
|---|---|---|---|---|---|
| % Accuracy | 67.3 | 68.1 | **70.2** | 68.7 | 67.8 |

We also create a pseudo-labeling baseline AVE+PL, where the base model's segment-level label predictions on the training data (pseudo-labels) are taken as ground truth for fully-supervised re-training. However, unlike AVE+LR, AVE+PL only marginally improves on AVE. The reason is while AVE+PL requires accurate *segment-level* predictions for re-training, AVE+LR only requires *video-level* predictions (see Fig. 2). The base model is more reliable for the latter since the loss during MIL training (Step1) is only applied at the *video level*. This validates the need for synthetic videos in our approach– we can estimate localized predictions from *video-level* outputs alone.

**Detection Threshold.** The hyperparameter $\tau \in (0, 1)$ in Eq. (11) is a measure of our trust in the model's predictions for the retained segments of the synthetic videos. We first obtain a candidate range for $\tau$ by performing Step2 on videos taken from the validation set and comparing our localized labels to the available ground truth. Tab. 4 shows the test accuracy of AVE+A+LR for different choices of $\tau$ in this range. The optimal value is $\tau = 0.05$. Note that we are using the SoftMax activation (not Sigmoid) (Eq. (6)) since AVEL assumes that only one event can occur at a given instant. Out of the 28 possible categories, the predicted one must receive a probability exceeding $1/28 \approx 0.036$ (*not* 0.5). Thus, our empirical value of 0.05 supports intuition.

### 5.4 Improving a different Base Model

To check if our method works with a different base model, we report performance using PSP (Zhou et al., 2021), a state-of-the-art architecture for AVEL, in Tab. 5. We did not expect our method to improve upon an already highly performant model, yet it significantly increases performance for both AVE and PSP on all the metrics considered. This shows that large performance gains may be possible by simply improving the training strategy, without architectural changes. We also show results for the challenging AVVP task (Tian et al., 2020) in Appendix A.1 and significantly outperform the baseline model in a setting where the *model architecture, dataset, and task* are different from AVEL.

### 5.5 Computational Cost

We take the number of forward passes per training example in Step1 (or Step2), $T_1 = (T - N)/s + 1$ as a measure of computational cost. Our method is $\mathcal{O}(T_1)$ times as expensive as training the base model alone, due to the auxiliary objective in Step1 and label refinement in Step2. Tab. 7 in Appendix A.2 compares performance for different costs. We find that performance is worst when the cost is lowest ($T_1 = 2$) since the labels would not be estimated at a fine enough resolution for meaningful re-training. The best performance was for a moderate cost of $T_1 = 4$.

Table 5: Performance for two different base models.

| Method | Accuracy | Wt. F1 | Non-AVE F1 (R/P) | AVE F1 (R/P) |
|---|---|---|---|---|
| AVE | 67.1 | 61.5 | 2.1 (1.1/25.8) | 73.7 (81.3/67.4) |
| AVE+A+LR | **70.2** | **68.6** | **32.3** (25.5/43.9) | **76.4** (79.9/73.2) |
| PSP | 70.0 | 64.6 | 5.9 (3.3/29.3) | 77.2 (84.7/70.9) |
| PSP+A+LR | **72.2** | **69.7** | **25.8** (18.2/44.3) | **79.0** (84.1/74.5) |

## 6 Conclusion

We presented a method that uses the predictive power of a decent base architecture for weakly supervised AVEL to produce temporally refined event labels for the training data. We introduced a novel auxiliary training objective that aids in the reliable generation of these labels. We showed how to re-train the base architecture using the generated labels. We then highlighted the issues with using a single metric to evaluate performance on AVEL. Finally, we carried out extensive evaluations and showed that our method outperforms several existing methods with no architectural novelty.

**Limitations and Future Work.**

1. Our label-refinement procedure is computationally expensive. Step2 and the auxiliary objective for Step1 require $T_1$ forward passes through the base model for each $V^{(i)}$, scaling linearly with video length $T$ (but all passes within a step can be parallelized).

2. Performance is sensitive to the threshold $\tau$ and requires precise tuning on a validation set since a wrong choice of $\tau$ means Step3 gets trained with incorrect labels under *stronger* supervision. Future work could directly use the predicted probabilities instead of binarizing them with a threshold. It could also explore training several iterations of our method, with a continually improving base model.

3. It is not clear how many related video-/image-based prediction tasks our method can generalize to. As an encouraging first step, we have provided results for AVVP, a video-based temporal- *and* modality-level prediction task (see Sec. 2) in Appendix A.1, and achieve large improvements on several metrics over the baseline model, which uses self- and cross-attention (Vaswani et al., 2017) instead of a Bi-LSTM. However, more evidence of adapting our method to related tasks would be useful in future work.

To conclude, we hope this paper will inspire future work to devise even better training strategies that push the limits of weakly-supervised performance.

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

# A  Appendix

## A.1  Performance on AVVP

Tab. 6 shows results on the challenging Audio-Visual Video Parsing task. We start with the base architecture `HAN` (Tian et al., 2020) and re-train it with our refined labels (`HAN+LR`), as discussed in the paper. `HAN+LR` outperforms `HAN` on almost all the metrics proposed by Tian et al. (2020) while requiring no changes to the base architecture.

Table 6: Performance comparison with the `HAN` baseline.

| Method | Audio | | Visual | | Audio-Visual | | Type@AV | | Event@AV | |
|---|---|---|---|---|---|---|---|---|---|---|
| | Seg. | Event | Seg. | Event | Seg. | Event | Seg. | Event | Seg. | Event |
| HAN | 60.1 | 51.3 | 52.9 | 48.9 | 48.9 | 43.0 | 54.0 | 47.7 | 55.4 | 48.0 |
| HAN+LR | 59.7 | **52.2** | **57.9** | **54.0** | **52.6** | **46.9** | **56.7** | **51.1** | **56.6** | **49.2** |

## A.2  Choices of $N$ and $s$

Tab. 7 shows results for different choices of the refinement window size $N$ and stride $s$. Reasonable choices must satisfy the following: (i) $s|(T - N)$ so *all* segments are covered by the window, (ii) the number of forward passes $T_1 = (T - N)/s + 1$ is small to reduce the computational cost, (iii) $N < \frac{T+1}{2}$ (see Sec. 4.3), and (iv) the window covers each segment a roughly equal number of times. We tune the detection threshold $\tau$ separately for each choice. We get the worst performance with $N = 5$ and $s = 5$ since the event labels here are not estimated at a fine-enough resolution (i.e., 1 for every 5 segments). Performance is comparable amongst the other choices.

Table 7: Performance of `AVE+A+LR` for different $N$ and $s$ choices. $T_1$ is a measure of the computational cost during training. All other numbers are percentages.

| Choice | Accuracy | Wt. F1 | Non-AVE F1 (R/P) | AVE F1 (R/P) | $T_1$ |
|---|---|---|---|---|---|
| $N = 2, s = 2$ | 69.5 | 68.7 | 40.1 (35.4/46.3) | 75.0 (76.9/73.2) | 5 |
| $N = 3, s = 1$ | 69.4 | 68.3 | 33.6 (27.5/43.3) | 75.5 (78.5/72.7) | 8 |
| $N = 4, s = 2$ | 70.2 | 68.6 | 32.3 (25.5/43.9) | 76.4 (79.9/73.2) | 4 |
| $N = 5, s = 5$ | 68.7 | 66.0 | 23.6 (16.1/43.9) | 75.0 (80.1/70.5) | 2 |

## A.3  Qualitative Results

We qualitatively compare the performance of our method (`AVE+A+LR`) with `PSP` in Figs. 5 to 14. We also include the ground truth for each example. As discussed in the paper, our method more accurately localizes the audio-visual events in addition to classifying them into known categories. Previous methods often fail to handle the localization task very well.

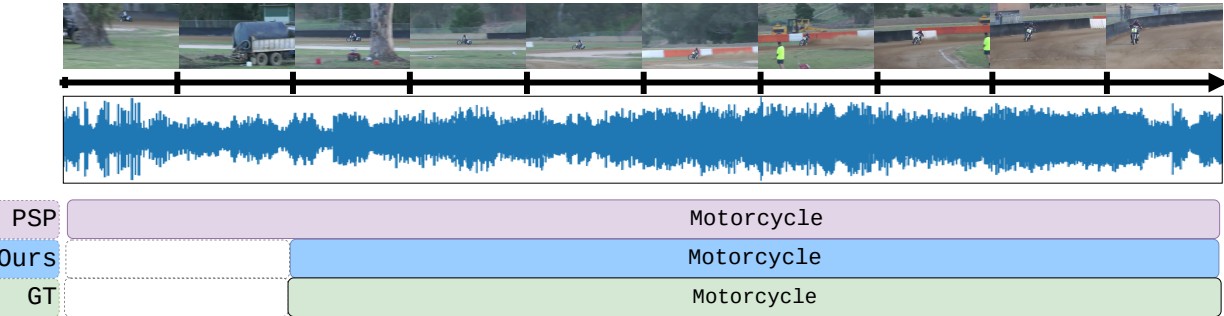

Figure 5: Motorcycle. Previous methods (e.g., PSP) often predict the video-level category for every frame, while our method can accurately localize the audio-visual event within the video.

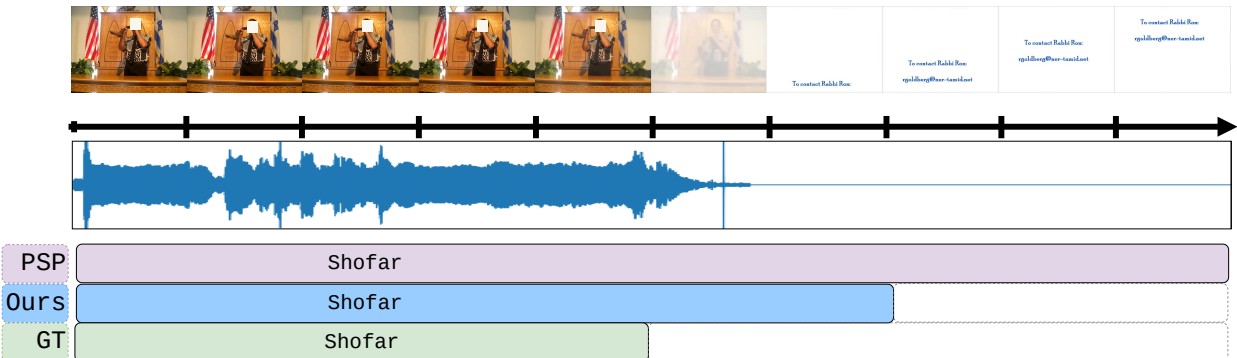

Figure 6: Shofar (instrument). While our method localizes the event better than the previous method, it incorrectly predicts the event during $[5, 7]$ seconds.

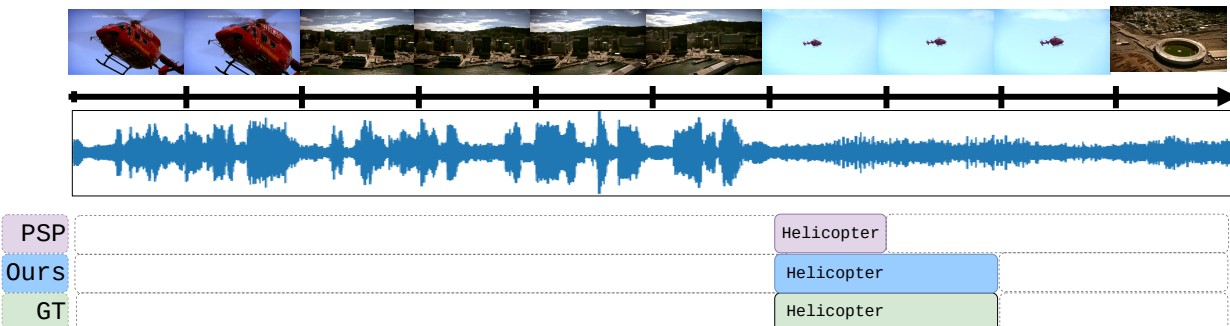

Figure 7: Helicopter.

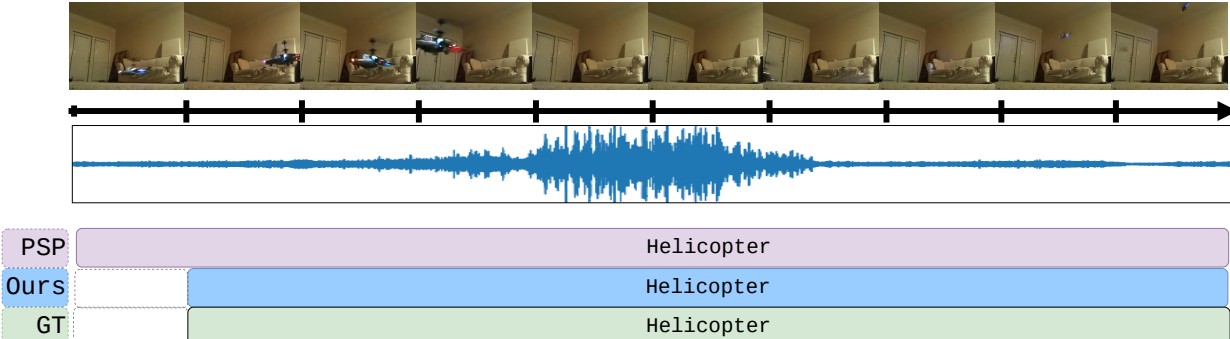

Figure 8: Helicopter.

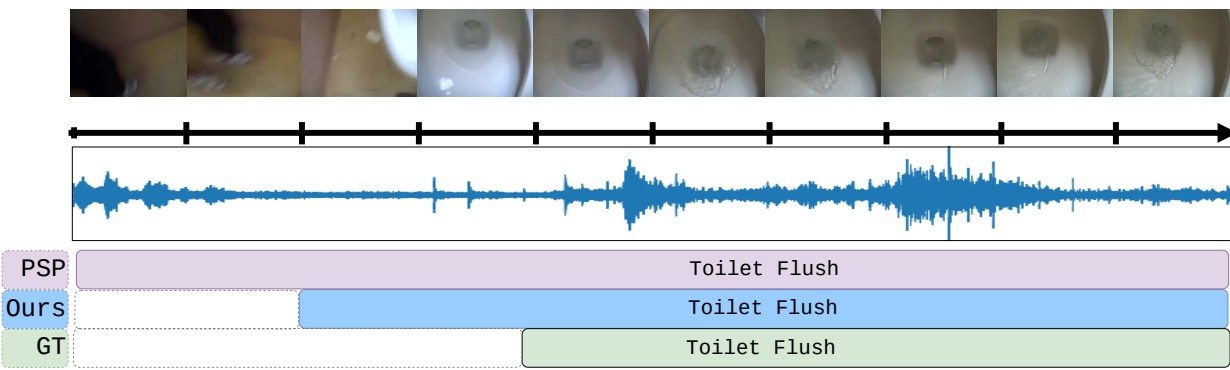

Figure 9: Toilet Flush. Our method incorrectly predicts the event during $[2, 4]$ seconds.

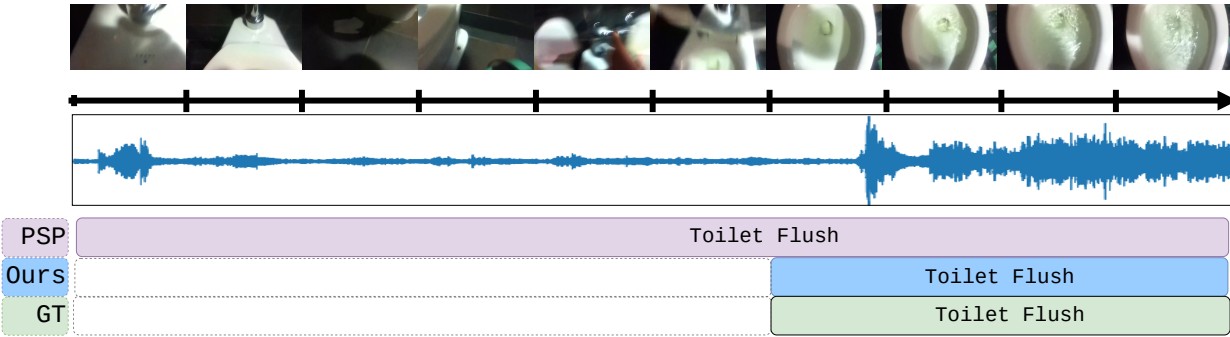

Figure 10: Toilet Flush.

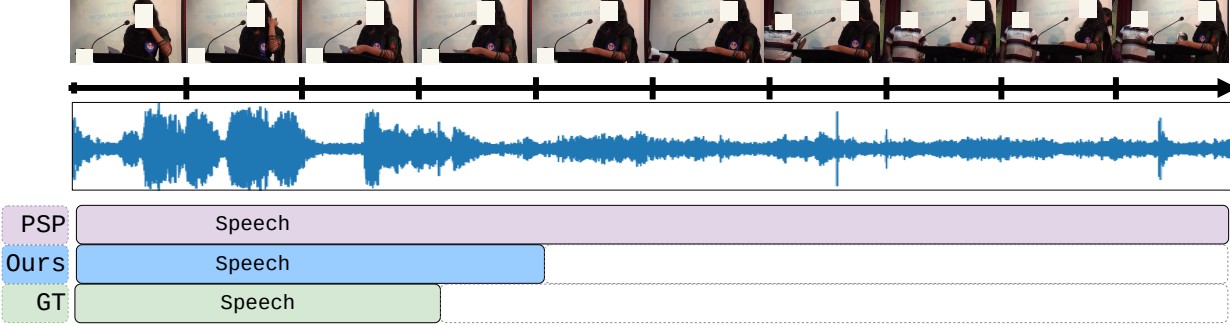

Figure 11: Woman Speaking.

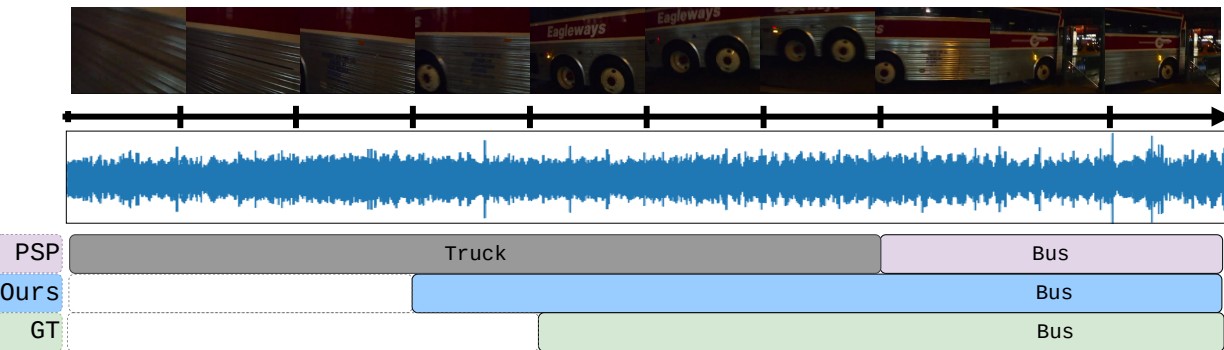

Figure 12: Bus. In addition to not localizing the event well, the previous method predicts an incorrect category (Truck) when no audio-visual event occurs in the video.

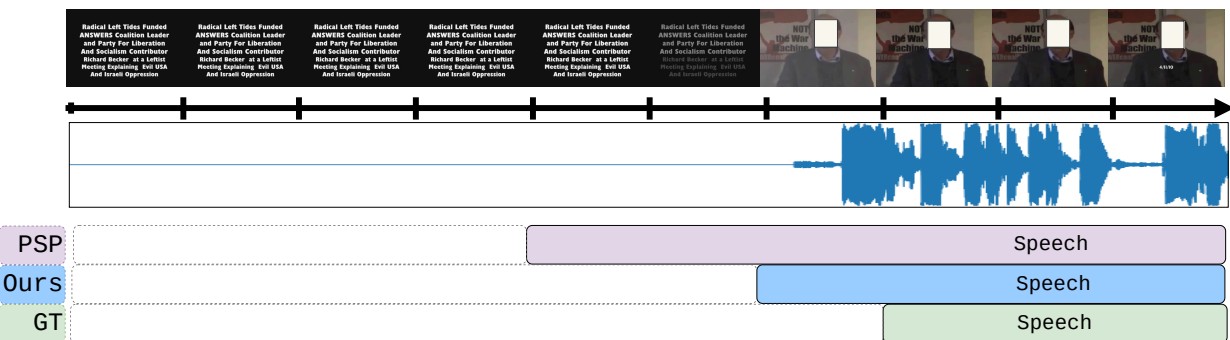

Figure 13: Man Speaking.

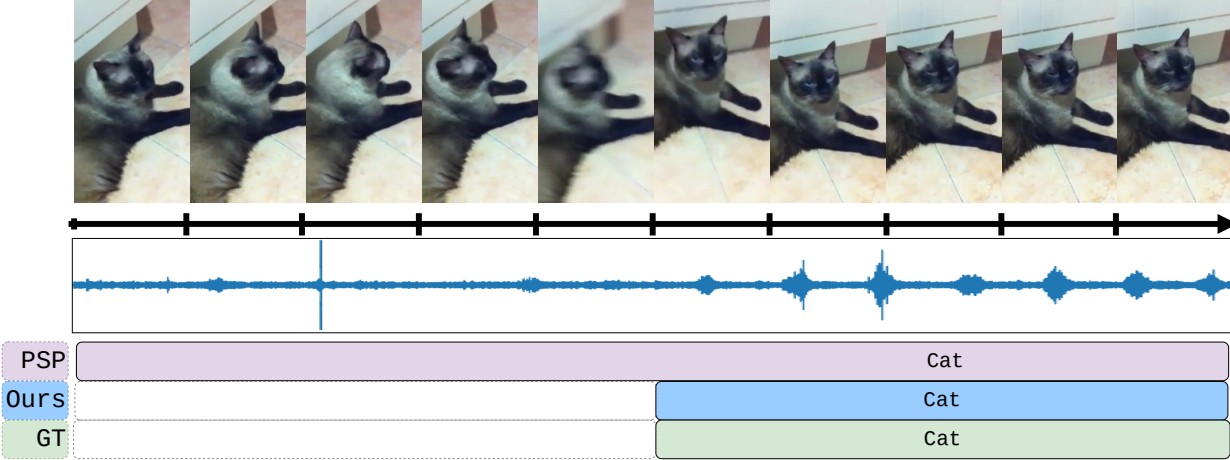

Figure 14: Cat.

