# OpenReview forum: "Temporal Label-Refinement for Weakly-Supervised Audio-Visual Event Localization"
_TMLR — Rejected by TMLR_

### Review · Reviewer_C9oG · 2023-11-26

**Summary Of Contributions:**

The authors tackle the Audio-Visual Event Localization problem. The key idea is a so-called ‘label-refinement’ approach, where a seed model is used to predict a slice of frames such that a synthetic augmented dataset could be curated, and the curated would gain feeding back to iteratively improve the seed model.

The authors also argue on the limitation of current widely used accuracy metrics, and propose improved approaches to evaluate AVEL.

**Audience:**

Yes

**Claims And Evidence:**

Yes

**Requested Changes:**

The authors could potentially consider testing on other datasets like UnAV-100, AudioSet, VGG, etc — To demonstrate both the effectiveness of their approaches in multiple widely used datasets, and also test further support their claim on accuracy on its own is infeasible.

The authors could consider comparing to more recent methods in their table 2. Also, in table 2, the authors could also consider listing the model complexity (e.g., number of trainable parameters, if or not using pre-trained components for each method that they compare with).

Should the authors consider comparing with other weakly supervised and/or augmentation approaches? The authors mentioned that their approach is similar to mixup; Should the authors compare with mixup, or say why mixup is not feasible for the problem?

**Strengths And Weaknesses:**

Strength:

— The proposed idea is simple but makes sense.

— The draft is overall easy to follow.

— The authors show clear evidence to support their claims: a) the existing metrics are problematic, and b) the proposed methods do improve upon some existing baselines. Further ablation studies are conducted to better understand the gain from the label refinement.



Weakness:

— The authors only work on the AVE dataset collected by Tian et al. (2018).

— The authors only demonstrated the issues of accuracy metric using a single dataset.

— The authors do not compare to most recent works (e.g., works published in 2022 and 2023).

— The authors do not compare to more traditional data augmentation or other weakly-supervised methods.

— The proposed core idea (see Figure 3) could be favoring recall after pooling, and be tailoring towards false acceptance.

---

> ### Comment · Editors_In_Chief · 2023-12-19
> **Response to Reviewer C9oG**
>
> Thank you for the review. We are happy you found our claims backed by evidence and our findings relevant to the TMLR audience. Here is our point-by-point response:
>
> **Summary of Contributions:** To make sure we are on the same page, label-refinement *does not* “predict a slice of frames” in a training video or curate a “synthetic augmented dataset”. It should instead be viewed as a *slice-level pseudo-label extraction* method for an existing dataset. As mentioned in the abstract, we use a base model to predict *pseudo labels* for each slice of frames in a training video, so that we can then re-train the model under stronger supervision from these labels. To enable extracting such pseudo labels, we temporarily replace all frames outside the slice with external content – forming a synthetic video – and feed this into the base model and get video-level predictions. From the video-level predictions for the synthetic video, we can guess what the labels are for just the slice in question. This is the label-refinement process, illustrated in Fig. 2, where we estimate pseudo labels for the slice in green (the estimated pseudo labels are shown with a tick).
>
> **Only used AVE dataset:** In Appendix A.1, we provide results for the challenging AVVP task and significantly outperform the baseline on the Look, Listen, and Parse (LLP) dataset. Moreover, our objective was to solve audio-visual event localization (AVEL), and the AVE dataset is the one used by all prior work (see Tab. 2). As for whether our method generalizes to related video-based prediction tasks, we leave that as future work, as discussed in Sec. 6.
>
> **Accuracy issue only for AVE dataset:** Yes, we are not claiming that accuracy is misleading in general - the issues with accuracy are indeed unique to the AVE dataset. However, all prior work has overlooked this, claiming “state-of-the-art” for AVEL while reporting misleading results. Ours is possibly the only work to solve AVEL at a reasonable level (see non-AVE scores in Tab. 2).
>
> **Compare to recent work:** Please let us know which specific work we should compare against.
>
> **Compare to traditional data augmentation:** As mentioned earlier in the response, label-refinement *is not* a data augmentation method. On the other hand, the auxiliary objective *can* be viewed as a data augmentation method while training the base model in step 1, but this is specifically to facilitate reliable label-refinement (inference on synthetic inputs) in step 2. For this, we implement the specific prior that across all *convex concatenations* of input videos, the base model can recover all the events occurring in them, as explained in Sec. 4.3. No traditional methods implement the prior we are looking for (since we are the first to propose temporal label-refinement), so do not make sense for our method. This also applies to Mixup regularization.
>
> **Compare to other weakly-supervised methods:** In Tab. 2, we compare against 8 other published weakly-supervised methods. In Tab. 3, we compare against naive pseudo labeling (AVE+PL) as a weakly-supervised method. In Appendix A.1, we compare with another published weakly-supervised method for the AVVP task.
>
> **Method encourages false positives:** We observe the opposite. Our method *reduces* false positives - see AVE precision (“AVE F1(P)”) in Tabs. 2 and 3, where our numbers are higher than all other methods. Moreover, as hypothesized in Sec. 4.3, our auxiliary objective helps reduce false negatives – see non-AVE precision (“Non-AVE F1(P)”) in Tab. 3, where AVE+A+LR gets a higher score than AVE+LR. We also want to clarify that our “core idea” (label-refinement/slice-level pseudo label extraction) is shown in Fig. 2, not Fig. 3. The latter shows the auxiliary objective we use for the base model in step 1 to prepare it for label-refinement/step 2.
>
> **Trainable params:** We will include these in Tab. 2 in the final version: \
> AVE = 1,335,246 \
> CMRA = 15,863,519 \
> PSP = 1,254,251 \
> Ours (AVE) = 1,335,246.

---

### Review · Reviewer_VPhC · 2023-12-05

**Summary Of Contributions:**

This paper addresses the audio-visual event localization (AVEL) task, defined as the classification and localization of events simultaneously visible and audible in a video under a weakly supervised setting (i.e., just the event presence or absence is known, but not its temporal localization). The proposed approach is based on a base model that is able to estimate pseudo-labels from the training data at a finer resolution, followed by the re-training of the model with this additional information. Such pseudo-labeling is done by building synthetic videos by mixing snippets of different videos and by designing an auxiliary objective loss to train the base model.

**Audience:**

Yes

**Broader Impact Concerns:**

I do not see any ethical implications in this work.

**Claims And Evidence:**

No

**Requested Changes:**

I'd like the authors to address the weaknesses reported above.

**Strengths And Weaknesses:**

Strengths
- The addressed task is relevant from a theoretical and practical standpoint, mitigating the need for accurate annotation.
- The method seems to have a certain level of originality.
- Results are state of the art, and nice ablation analysis.

Weaknesses
1. The definition of AVEL reads as a simultaneous audio AND visual event, but it sounds a bit artificial to me: in fact, events in a generic video can be detected as deriving from audio or video or both, with different weights, so focusing on those simultaneously audio and visual seems to me an ad hoc sub-task.
2. The proposed method builds on top of the work of the model of Tian et al. (2018), but more details are needed to fully understand it without resorting to read the original paper. For example, MIL is not so popular nowadays and should be better illustrated.
3. The method's description does not proceed smoothly, making understanding some stages of the method unclear. The overall procedure is not well justified, in particular, the logical steps for generating synthetic video generation are unclear, as well as how the finer subdivision in segments is obtained (and how much it affects the performance). Likely, this is also due to the notation, which is a bit complex to follow precisely, e.g., eqs. 7 and 8 should be better explained.
4. Similar considerations are valid for the introduction of the auxiliary training objective, and in the end the total loss is missing. Overall, the proposed method appears quite empirical with no theoretical ground, and also the general intuition is hard to catch.
5. While the achieved results are better than the state-of-the-art algorithms, this does not hold true for all the reported evaluation metrics, and comments are missing in this respect. Moreover, among such metrics, I would have expected at least a measure estimating the precision of the temporal localization (something like IoU of the estimated time window wrt ground-truth window of the AVE), but I did not find it.
6. The figures representing the models (no. 2 and 3) are not much informative, and do not help a better understanding of the approach.

---

> ### Author Response · Authors · 2024-01-09
> **Response to Reviewer VPhC (1/2)**
>
> Thank you for the review. We are happy you found our findings relevant to the TMLR audience. After reading our response, **please let us know what claims we have made in the paper that you feel have not been backed by evidence**. We are happy to revise our claims in this case. Here is our point-by-point response:
>
> **AVEL is just a sub-task:** Agreed, events can indeed occur in either modality, so AVEL is not useful in practice. However, this does not mean it is an easy task to solve – from Tabs. 1 and 2, we see that none of the 8 previous methods compared can actually solve this task in the weakly-supervised setting (they essentially make the same prediction for all frames, as explained in Sec. 5.1). Thus, AVEL is still an interesting problem from a weakly-supervised methods standpoint. We will include this in the introduction of our final version.\
> Also, in Appendix A.1, we evaluate our approach on the challenging audio-visual video parsing (AVVP) task, where events *do occur* asynchronously across modalities. Here, our method significantly outperforms the baseline on several metrics. As for whether we can generalize to related, more practical video-based prediction tasks, we leave that as future work, as discussed in Sec. 6.
>
> **More details for Tian et al.:** We agree this would be useful for clarity. We will add an overview figure for the base model in the appendix in the final version and direct readers there (we did not include it in the paper since we wanted the focus to be on our training pipeline, not the architecture).
>
> **Method description unclear:**
> - **How to generate synthetic videos:** We think this is clear from Fig. 2 and in the text above Eq. (9). To estimate slice-level pseudo labels for the green slice (window) in video V(1) in Fig. 2, we replace all frames outside the slice with those from video V(2). This is the synthetic video, which we feed into the base model and extract video-level predictions for (see curly braces on the right). We use these video-level predictions to infer pseudo labels for just the green slice from V(1), as explained in the caption.
> - **How are videos split into segments:** The 10-second videos have already been split into 10 segments each (with 16 frames per segment) and their extracted features are provided in the AVEL dataset, so this is not under our control.
> - **Notation complex:** We think every term appearing in Eqs. (7) and (8) has been clearly defined in Sec. 4.2, in the paragraph called “Notation”.
>
> **Total loss missing:** We completely agree that this would make understanding the method much easier, so will definitely include it in the final version.
> In Step 1 (under “Implementation Steps” in Sec. 4.2), we train the base model with the MIL loss and the auxiliary objective, so the loss is $L_{\text{MIL}} + L_{\text{A}}$ (see Eq. (18)). In Step 2, we use the pre-trained model from Step 1 to extract slice-level pseudo labels - there is no training here, and hence no loss function. In Step 3, we re-train the base model under stronger (pseudo-label) supervision from the labels inferred in Step 2. So, the loss here is $L_\text{LR}$ (Eq. (13)).
>
> **Not better on all metrics:** Agreed, while we outperform compared methods (see Tab. 2) on accuracy, overall F1 score, average non-AVE recall, non-AVE precision, and AVE precision, we get lower performance on average AVE recall. Since we significantly outperform previous methods on non-AVE metrics (and hence at temporal localization), the reason for the lower score is that the method may make a few misclassifications for certain events (e.g., when the visual stream contains more than one event). This would also explain why our improvement in overall accuracy is not as large as in other metrics. Will add it to the final version.
>
> **No IoU metric:** IoU is essentially equivalent to the overall F1 score (reported in Tab. 2, column 2) since we are dealing with a fixed (discrete) number of segments per video. To see this, note that IoU = TP/(TP+FP+FN), where TP, FP, and FN are true positives, false positives, and false negatives, respectively. But, precision = TP/(TP+FP) and recall = TP/(TP+FN). So, F1 = TP/(TP + (FP+FN)/2), which is similar to IoU. Moreover, reporting precisions/recalls for AVE classification and non-AVE detection gives a more complete picture of performance and helps expose the oversimplification of the task by previous methods, as explained in Sec. 5.1 and Tab. 1.
>
> **Figures not informative:** We think Fig. 2 and the accompanying caption illustrate our label-refinement idea clearly. We have explained Fig. 2 further in the “method description” part of this rebuttal. If Fig. 2 is still not clear, **please let us know what should be improved**, and we will include your suggestions in the final version. We agree that Fig. 3 should be improved – in the final version, we will update it with frames from actual videos, like in Fig. 2.

---

> ### Author Response · Authors · 2024-01-09
> **Response to Reviewer VPhC (2/2)**
>
> **Intuition hard to grasp:** We are happy to explain the three steps of our pipeline in simpler terms below. **Please read through this entirely**:
>
> **(1) Step 1:** We train a base model using video-level labels, just as explained in the original AVEL paper. However, we also add an extra auxiliary loss here, as explained later.
>
> **(2) Step 2 (label-refinement):** Here, we will use this pre-trained model to estimate *pseudo labels* for *slices* of frames in each training video, as explained below. "Slice" here means N consecutive frames in a video (the video has T frames and N<T).
>
> - **Motivation:** If we could estimate accurate pseudo labels for slices with N=1, i.e., at the frame level, we could thereafter simply train a model against these frame-level labels, recovering the fully-supervised setting. However, a weakly-supervised pre-trained model cannot be expected to estimate frame-level labels accurately, so this idea would not work.
> Instead, we try to estimate accurate pseudo labels for slices with N>1, for each training video. Once we do this, we can thereafter train a model against these *slice-level* labels, resulting in a setting that is *in between* full supervision (having labels for slices with N=1) and weak supervision (having labels for slices with N=T, or at the video level).
>
> - **Method:** Fig. 2 shows our method to extract such slice-level pseudo labels. Here, we are estimating slice-level pseudo labels for a particular slice in V(1), shown in green. To do this, we find a V(2) with no common events and replace everything in V(1) outside the slice with content from V(2). This is the synthetic video. Now, since the pre-trained model in Step 1 was trained with a *video-level* loss, it can be expected to act as an accurate *video-level* predictor. So, we get its *video-level* predictions on the synthetic video, shown within the curly braces on the right. Since the bus and the dog are present in the synthetic video but the person is not (please zoom in), the model's video-level predictions are above the threshold for Bus (0.5) and Dog (0.4), but not for Person (0.01). Thus, Person can be discarded as a possible slice-level label for the green slice. Moreover, Bus can also be discarded as a possible slice-level label since Bus is not present in V(1), so definitely could not have occurred within the slice! We are finally left with the set {Dog}, which is our prediction for the slice-level labels for the green slice. We repeat this for all slices (moving at a stride $s$) in the video, estimating and storing slice-level pseudo labels for each. We do this for all videos in the training data.\
> This method allows us to exploit the pre-trained model's *video-level* predictive ability to predict pseudo labels for slices that are more localized in the video. We hope it is now clear why synthetic videos are necessary for slice-level pseudo-label extraction. This is further validated by the ablations in Sec. 5.3.
>
> **(3) Step 3:** Now that we have slice-level pseudo labels for all slices in all training videos, we train a model from scratch against these labels. This completes the pipeline.
>
> **(AO) Auxiliary Objective:** This can be viewed as a data augmentation method while training the base model in Step 1, to encourage the pre-trained model's predictions on synthetic inputs to become more reliable, thus preparing the model for Step 2. We are happy to explain AO further and how we integrate it into Step 1 once you are satisfied with our explanation on label-refinement, which is the core idea of the paper.

---

> > ### Author Response · Authors · 2024-01-30
> > **Following up with Reviewer VPhC**
> >
> > It has been 2 weeks since our rebuttal but we have not yet received a response from you about claims made in the paper that you felt were not backed by evidence.
> >
> > If you have changed your opinion, we request you to update "Claims and Evidence" from "No" to "Yes". Otherwise, please let us know which claims we should reduce.

---

> > > ### Author Response · Authors · 2024-02-10
> > > **Following up with Reviewer VPhC**
> > >
> > > Following up again.

---

### Review · Reviewer_ZxBm · 2024-01-07

**Summary Of Contributions:**

This paper studies a weakly supervised setting of the classical Audio-Visual Event Localization. The authors assume that only video-level event labels (the presence/absence) are available, and this is no temporal location as supervision for training. To solve this new setting, they proposed to use a pre-trained network to predict the AVE location in time. Then, they use the predicted pseudo labels to train the model. Experiments on popular datasets validate the effectiveness of the proposed method.

**Audience:**

Yes

**Claims And Evidence:**

Yes

**Requested Changes:**

See the above

**Strengths And Weaknesses:**

__Strengths:__

-The problem setting is interesting and might be useful in some large-scale cases. It is indeed much easier to just provide the video-level annotation than the frame-to-frame annotation.

-Experiment results are reasonably good compared with baseline methods on several top conferences.

__Weaknesses:__

-This is more like a CV paper and does not very match well with an ML journal TMLR. I would encourage the author to submit it to some CV journal.

-The novelty and contribution are somewhat limited. The main contribution of this paper is applying a pre-trained network to the existing AVE method. There is no novelty in the methodological aspect.

-I do not think the idea of using a pre-trained network to predict the temporal location is technically sound for a weakly supervised setting. Actually, to obtain such a reliable network, we may still provide lots of annotations besides the video-level label itself.

-The experimental comparison is somewhat unfair. Most of the compared methods are not designed for such a weakly supervised case, I do not think that directly applying them in this new setting can really validate the superiority of the proposed method. Maybe we should consider to compare with some other weakly supervised learning approaches.

---

> ### Author Response · Authors · 2024-01-17
> **Response to Reviewer ZxBm**
>
> Thank you for the review. We are happy you found our claims backed by evidence and our findings relevant to the TMLR audience. Here is our point-by-point response:
>
> **Method just uses a pre-trained network to get pseudo labels:** This is a serious mischaracterization of our contributions. We are happy to clarify:
> - Simply pseudo-labeling with a pre-trained model and re-training **does not** work for AVEL (see AVE+PL, Tab. 3). We explain why in Sec. 5.3 - while the pseudo labeling baseline AVE+PL requires accurate *frame-level* predictions from the pre-trained model for re-training, our label-refinement approach (AVE+LR) only requires accurate *video-level* predictions. This is a weaker requirement for the pre-trained model since it was only provided with *video-level* labels when training. Thus, our method is a way to utilize the pre-trained model’s video-level predictive ability to extract pseudo labels for each slice of frames in a training video. How do we achieve this? With the help of carefully designed synthetic videos, as shown in Fig. 2.
> - We also do not predict pseudo labels for each *frame* in a training video with our method – this would be computationally expensive. We instead predict pseudo labels for each *slice* of $N$ frames, moving at a stride $s$ (we call this “temporal label-refinement”). It is not obvious how to re-train the model using such *slice-level* pseudo labels - we show how to do this in Sec. 4.2 (implementation step 3).
> - While the pre-trained model is a better video-level predictor, it still hasn’t seen synthetic videos when training. For this, we include an auxiliary loss function when training it, as explained in Sec. 4.3, and show that it reduces false-negative predictions (as expected) in the ablation study in Sec. 5.3, improving overall performance (see Tab. 3).
> - In Sec. 5.1 (Tab. 1), we show that previous methods have used an insufficient evaluation method, and propose several new metrics for a more complete evaluation. The non-AVE metrics in Tab. 2 show that these methods do not solve the AVEL task at all, essentially treating it as a video classification problem. To our knowledge, our work is the only one to actually solve AVEL at a reasonable level.
>
> **Paper not suited to TMLR:** We think this objection is invalid. Our submission is within scope (see “scope” at https://www.jmlr.org/tmlr/editorial-policies.html) since it falls under “accounts of applications of existing techniques that shed light on the strengths and weaknesses of the methods” and “reproducibility studies of previously published results or claims”.
> Moreover, there is no requirement that TMLR papers do not look/read “like a CV paper” in the author guidelines (https://www.jmlr.org/tmlr/author-guide.html).
>
> **Novelty limited:** We think there are at least three novel ideas in our paper **(if you disagree, please let us know which specific works have similar ideas)**:
> - Slice-level pseudo-label extraction (temporal label-refinement) in a sliding window fashion with the help of special synthetic videos.
> - Re-training the model in a middle-ground setting between strong and weak supervision (Sec. 4.2) using slice-level pseudo labels.
> - The auxiliary loss for the base model to facilitate reliable inference on synthetic inputs.
>
> **Using a pre-trained network is not sound:** Yes, using a pre-trained network to directly predict pseudo labels does not work (see AVE+PL, Tab. 3), as explained in Sec. 5.3. This is precisely why we take a different approach – creating synthetic videos and using *video-level* predictions to estimate pseudo labels for *slices* of frames.
>
> **Pre-trained model unreliable for pseudo labels:** Yes, that is exactly why we think our contributions are valuable. If we want the pre-trained model to generate reliable pseudo labels directly, we would need to train it under full supervision, which we do not have access to. This is the motivation for our approach (see “Motivation” under Sec. 4.2), where we create a middle-ground (slice-level) setting between strong (frame-level) and weak (video-level) supervision, instead of a fully-supervised setting with (unreliable) frame-level pseudo labels.
>
> **Comparisons unfair:** The 8 other published methods we compare against in Tab. 2 all propose architectural/feature-aggregation improvements aimed at *both* the fully-supervised and weakly-supervised settings in AVEL. This is *not* a new setting for previous methods. In Tab. 3, we compare against naive pseudo-labeling (AVE+PL) as a weakly-supervised method. In Appendix A.1, we compare with another published weakly-supervised method for the AVVP task. Moreover, unlike these methods, we fix the architecture to the very first baseline (Tian et al., 2018), and yet outperform them with a carefully-designed training strategy. **If you disagree, please let us know which specific works we should compare against.**

---

### Decision · Action_Editor_jpDM · 2024-03-03

**Recommendation:** Reject

**Comment:**

The reviewers were not convinced by the authors response. The manuscript is not ready for publication in TMLR, lacking sufficient justification of the methods, and sufficient experiment comparisons in terms of both recent methods and other datasets (see Claims & Evidence above).

The paper may find a more appropriate audience at a computer vision journal or conference.

**Audience:**

The paper involves empirical work in computer vision, so the audience will be limited.
Meanwhile, the rationale and justifications of the method are not sufficient, which could further limit the interest of the paper.

**Claims And Evidence:**

- The rationale and justifications of the method are not sufficient.
- There are more recent works in AVE that should be compared to make better claims about the method. From reviewer C9oG "There are many new works on AVE that may not be tested on AVEL. For example, the AVE-CLIP, and there should be many more by checking the citation of Tian 2018 which are from 2022 till now."
- Newer datasets could also be tested to improve the claims. From Reviewer C9oG "The authors are claiming that the issue on accuracy is on AVE in general, and the authors’ method works better in the new metric. I would request the authors to try some other newer trending datasets like UnAV-100. The authors’ response is that AVEL is the only dataset that all prior works are all working on."